

# Average symmetry protected higher-order topological amorphous insulators

**Yu-Liang Tao[1], Jiong-Hao Wang[1] and Yong Xu[1,2]⋆**

**1** Center for Quantum Information, IIIS, Tsinghua University,
Beijing 100084, People's Republic of China
**2** Hefei National Laboratory, Hefei 230088, People's Republic of China

⋆ yongxuphy@tsinghua.edu.cn

## Abstract

While topological phases have been extensively studied in amorphous systems in recent years, it remains unclear whether the random nature of amorphous materials can give rise to higher-order topological phases that have no crystalline counterparts. Here we theoretically demonstrate the existence of higher-order topological insulators in two-dimensional amorphous systems that can host more than six corner modes, such as eight or twelve corner modes. Although individual sample configuration lacks crystalline symmetry, we find that an ensemble of all configurations exhibits an average crystalline symmetry that provides protection for the new topological phases. To characterize the topological phases, we construct two topological invariants. Even though the bulk energy gap in the topological phase vanishes in the thermodynamic limit, we show that the bulk states near zero energy are localized, as supported by the level-spacing statistics and inverse participation ratio. Our findings open an avenue for exploring average symmetry protected higher-order topological phases in amorphous systems without crystalline counterparts.



## 1 Introduction

Although topological physics is mainly established in crystalline solids with translational symmetry, there has been a growing interest in studying it in non-crystalline materials, including amorphous materials [1–25] and quasicrystals [26–38], given their prevalence in condensed matter systems. Unlike crystalline systems that permit only two-fold, three-fold, four-fold or six-fold rotations, quasicrystals allow for a broader range of rotational symmetries, including five-fold or eight-fold rotations. Surprisingly, such symmetries enable quasicrystals to support new higher-order topological phases in two dimensions without crystalline counterparts [30,31,37]. Here, higher-order topological phases refer to topological phases that harbor gapless edge states at $(n-m)$-dimensional boundaries ($1 < m \leq n$) for an $n$-dimensional system [39–54].

Amorphous materials are another critical category of non-crystalline materials that fundamentally differ from quasicrystals. Unlike quasicrystals, amorphous lattices do not exhibit rotational symmetry and cannot be obtained by a projection from a higher-dimensional crystal. Previous studies have revealed that higher-order topological phases can exist in amorphous lattices, but they do not require protection by rotational symmetry [55,56]. The lack of rotational symmetry might suggest that amorphous systems lack higher-order topological phases without crystalline counterparts. However, although each random configuration of amorphous materials does not display any rotational symmetry, an ensemble of all possible configurations may exhibit an average symmetry [57–59]. For example, an ensemble of all randomly distributed sites in a cubic box preserves an average four-fold rotational symmetry [56]. Therefore, it is natural to ask whether the random arrangements in amorphous materials can create average crystalline symmetries that do not exist in crystalline materials and potentially generate higher-order topological phases beyond the crystalline ones.

In this work, we theoretically demonstrate the existence of higher-order topological insulators in two-dimensional (2D) random lattices protected by average crystalline symmetries. Such topological phases cannot exist in crystalline systems. For instance, we find that, through constructing and exploring a model Hamiltonian in a regular octagon, an ensemble of Hamiltonians in random lattices respects an average $C_8 M$ symmetry, although each sample does not preserve the $C_8 M$ symmetry. Here $C_8$ is the eight-fold rotational operator, and $M$ is an in-plane mirror operator. Such an average symmetry leads to an Anderson insulator with higher-order topology hosting eight zero-energy corner modes without crystalline equivalents (see Fig. 1). We further show that the topological phase is characterized by a $\mathbb{Z}_2$ topological invariant based on the average $C_p M$ symmetry. Notably, the quadruple moment is frequently used to characterize the topology of quadrupole topological insulators that support four corner modes [60–63], but it is not suitable for diagnosing the topology in our case where eight or more corner modes are present. To address this, we extend the conventional quadrupole moment to build a topological invariant capable of characterizing our model that accommodates $p$ corner modes for any positive integer $p$ that is a multiple of four. We finally illustrate the existence of a higher-order topological phase supporting twelve corner modes protected by the average $C_{12} M$ symmetry in a regular dodecagon.

## 2 Model Hamiltonian

To demonstrate the existence of average symmetry protected higher-order topological amorphous insulators, we consider the following tight-binding model on a 2D completely random lattice in a regular $p$-gon,

$$\hat{H} = \sum_{\boldsymbol{r}} \left[ m_z \hat{c}_{\boldsymbol{r}}^{\dagger} \tau_z \sigma_0 \hat{c}_{\boldsymbol{r}} + \sum_{\boldsymbol{d}} \hat{c}_{\boldsymbol{r}+\boldsymbol{d}}^{\dagger} T(\boldsymbol{d}) \hat{c}_{\boldsymbol{r}} \right], \tag{1}$$

where $\hat{c}_{\boldsymbol{r}}^{\dagger} = (\hat{c}_{\boldsymbol{r},1}^{\dagger}, \hat{c}_{\boldsymbol{r},2}^{\dagger}, \hat{c}_{\boldsymbol{r},3}^{\dagger}, \hat{c}_{\boldsymbol{r},4}^{\dagger})$ with $\hat{c}_{\boldsymbol{r},j}^{\dagger}$ ($\hat{c}_{\boldsymbol{r},j}$) creating (annihilating) a fermion of the $j$th component at the position $\boldsymbol{r}$, which is randomly distributed in a regular $p$-gon with the width $L$ [see Fig. 1(a) for the case with $p = 8$]. $\tau_{\nu}$ and $\sigma_{\nu}$ with $\nu = 0, x, y, z$ denote Pauli matrices that act on the internal degrees of freedom. In the above Hamiltonian, $m_z$ is the mass term, and

$$T(\boldsymbol{d}) = f(d)[t_0 \tau_z \sigma_0 + it_1(\cos\theta\,\tau_x\sigma_x + \sin\theta\,\tau_x\sigma_y) + g\cos(p\theta/2)\tau_y\sigma_0]/2, \tag{2}$$

is the hopping matrix from the site $\boldsymbol{r}$ to the site $\boldsymbol{r} + \boldsymbol{d}$, where $(d, \theta)$ are the polar coordinates of the vector $\boldsymbol{d}$. This Hamiltonian can be derived through a unitary transformation of the Hamiltonian introduced in Refs. [30, 31]. To mimic real material scenarios, we consider the relative hopping strength $f(d)$ that decays exponentially with the distance $d$, i.e., $f(d) = \Theta(R_c - d)e^{-\lambda(d/d_0 - 1)}$. Here, $\Theta(R_c - d)$ is the step function with $R_c$ being the cutoff distance so that hoppings for $d > R_c$ are neglected, and $\lambda$ denotes the strength of the decay. To make sure that the Hamiltonian is Hermitian, we consider $p$ that is an integer multiple of four. For simplicity, we set $t_0 = t_1 = 1$ and $d_0 = 1$ as the units of energy and length, respectively. In the following, we will set the system parameters $\lambda = 2$ and $R_c = 6$ in $f(d)$. In numerical calculations, we randomly place $N$ sites in a regular $p$-gon; each site has hard-core radius of $r_h$ for more realistic considerations (here we set $r_h = 0.2$) so that the distance between two sites cannot be less than $2r_h$. $N$ is given by $N = \lceil \rho A \rceil$, where $\lceil \rceil$ is the ceiling function which returns the nearest integer towards infinity, $\rho$ is the density of the system (we here set $\rho = 1$ without loss of generality), and $A$ is the area of the regular polygon.

We now show that an ensemble $\mathcal{E}_L$ of all sample configurations of randomly distributed lattice sites respects an average $p$-fold rotational ($C_p$) symmetry. For one sample configuration $\mathcal{R}_0$ consisting of positions of all lattice sites [see Fig. 1(a) when $p = 8$], it clearly does not respect the $C_p$ symmetry since $\mathcal{R}_0 \neq \mathcal{R}_1$ where $\mathcal{R}_n = (D_{C_p})^n \mathcal{R}_0 \equiv \{(D_{C_p})^n \boldsymbol{r} : \boldsymbol{r} \in \mathcal{R}_0\}$ with $n = 1, \ldots, p-1$. Here, $D_{C_p}$ is an operator that rotates the position vector $\boldsymbol{r}$ counterclockwise about the origin by an angle $2\pi/p$. However, for the statistical ensemble $\mathcal{E}_L$ in random lattice systems, $\mathcal{R}_1, \ldots, \mathcal{R}_{p-1}$ and $\mathcal{R}_0$ appear in the ensemble with the same probability, indicating that the ensemble respects the $p$-fold rotational symmetry on average.

In the following, we will demonstrate that the Hamiltonian in Eq. (1) respects an average crystalline symmetry on the ensemble of random lattices. For clarity, we write $\hat{H} = \hat{c}^{\dagger} H \hat{c}$ where $H$ is the first-quantization Hamiltonian. When $g = 0$, the system respects particle-hole symmetry $\Xi = \tau_x \sigma_x \kappa$ ($\kappa$ is the complex conjugate operator), i.e., $\Xi H \Xi^{-1} = -H$, time-reversal symmetry $T = i\sigma_y \kappa$, i.e., $THT^{-1} = H$ and chiral symmetry $\Gamma = \tau_x \sigma_z$, i.e., $\Gamma H \Gamma^{-1} = -H$. Since $T^2 = -1$ and $\Xi^2 = \Gamma^2 = 1$, the system belongs to the class DIII characterized by a $\mathbb{Z}_2$ topological invariant so that its nontrivial phase exhibits helical edge modes [64–66]. Because a random lattice configuration $\mathcal{R}_0$ does not have the $C_p$ symmetry, the Hamiltonian $H(\mathcal{R}_0)$ on the lattice generically does not respect the symmetry, that is, $C_p H(\mathcal{R}_0) C_p^{-1} = H(D_{C_p} \mathcal{R}_0) \neq H(\mathcal{R}_0)$ where $C_p = \tau_0 e^{-i\frac{\pi}{p}\sigma_z} R_p$ with $R_p |\boldsymbol{r}\,\alpha\rangle \equiv |D_{C_p}(\boldsymbol{r})\alpha\rangle$. However, if we consider an ensemble $\mathcal{E}_H$ of the Hamiltonians on the lattice ensemble $\mathcal{E}_L$, $\mathcal{E}_H \equiv \{H(\mathcal{R}) : \mathcal{R} \in \mathcal{E}_L\}$, then there emerges an average $C_p$ symmetry in the Hamiltonian ensemble because $H(\mathcal{R})$ and its symmetry conjugate partner $C_p H(\mathcal{R}) C_p^{-1} = H(D_{C_p} \mathcal{R})$ occurs in the ensemble with the same probability.

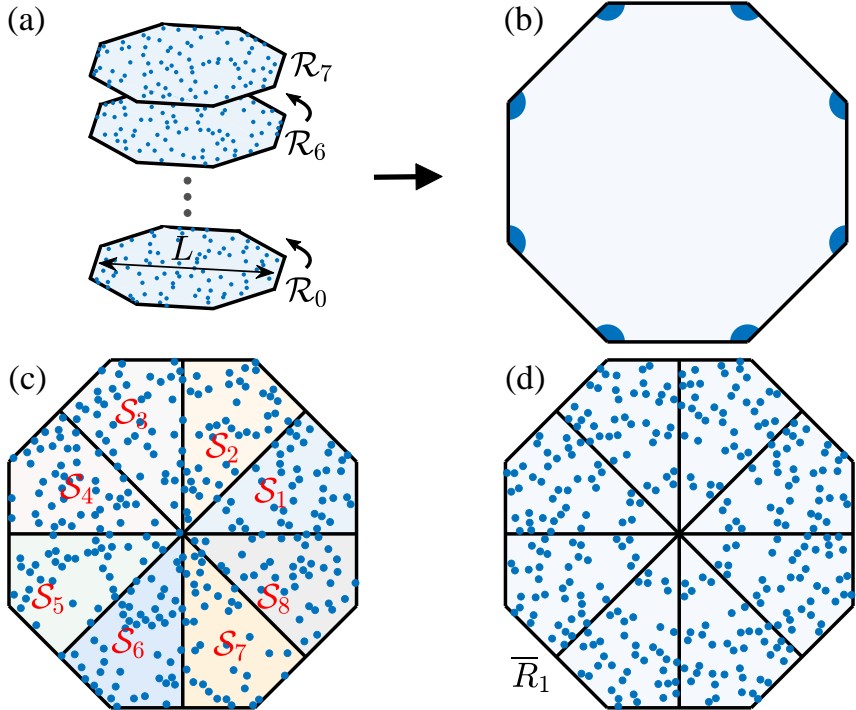

Figure 1: (a) Schematics of a lattice ensemble containing a random configuration $\mathcal{R}_0$ and the corresponding symmetry partners $\mathcal{R}_n = (D_{C_p})^n \mathcal{R}_0$ with $n = 1, \ldots, p-1$ obtained by rotating all the sites in $\mathcal{R}_0$ by an angle $2n\pi/p$ for $p = 8$. Such an ensemble leads to the average $C_8 T$ and $C_8 M$ symmetry, ensuring the existence of eight corner modes manifesting in the local density of states (DOS) as schematically shown in (b). (c)-(d) Schematics of how to change the lattice structure to the ones with eight-fold rotational symmetry so that the topological invariant $\chi$ can be evaluated. We first divide the lattice into eight sectors, each of which is labeled as $\mathcal{S}_j$ with $j = 1, \ldots, 8$ [see (c)]. We then reconstruct a new set of sites $\overline{R}_j$ by rotating the points in the $\mathcal{S}_j$ so that $\overline{R}_j = \{(D_{C_p})^n \mathcal{S}_j : n = 0, 1, \ldots, p-1\}$ [e.g., (d) displays the configuration of $\overline{R}_1$].

To generate the higher-order topological phase, we need to add the term $g \cos(p\theta/2)\tau_y \sigma_0$ to open the energy gap of the helical modes at boundaries by breaking the time-reversal symmetry. In this case, $C_p H(\mathcal{R}) C_p^{-1} \neq H(D_{C_p} \mathcal{R})$, and thus the average $C_p$ symmetry is broken. However, there arise new average symmetries $C_p T$ and $C_p M$ given that $(C_p T) H(\mathcal{R})(C_p T)^{-1} = (C_p M) H(\mathcal{R})(C_p M)^{-1} = H(D_{C_p} \mathcal{R})$ where $M = \tau_z \sigma_z$. Because of the average symmetries, if there exists a zero-energy corner mode $|\psi_c\rangle$ near the position $\boldsymbol{r}$ in the Hamiltonian $H(\mathcal{R})$, then there must exist a zero-energy corner mode $C_p T |\psi_c\rangle$ near the position $D_{C_p} \boldsymbol{r}$ in the Hamiltonian $H(D_{C_p} \mathcal{R})$. As a consequence, the configuration averaged local density of states at zero energy over the ensemble $\mathcal{E}_H$ must exhibit the $C_p$ symmetry as schematically shown in Fig. 1(b).

## 3 Topological invariants and topological property

To characterize the topological property of the system, we will build two topological invariants. For the first one, in order to ensure that the topological invariant is well defined, we divide each sample configuration into $p$ sectors as illustrated in Fig. 1(c) and use $\mathcal{S}_j$ with $j = 1, \ldots, p$ to describe the set of positions of lattice sites in each sector. We then use each position set $\mathcal{S}_j$ to generate a new lattice configuration $\overline{R}_j$ in the entire $p$-gon by rotating the points in the $\mathcal{S}_j$

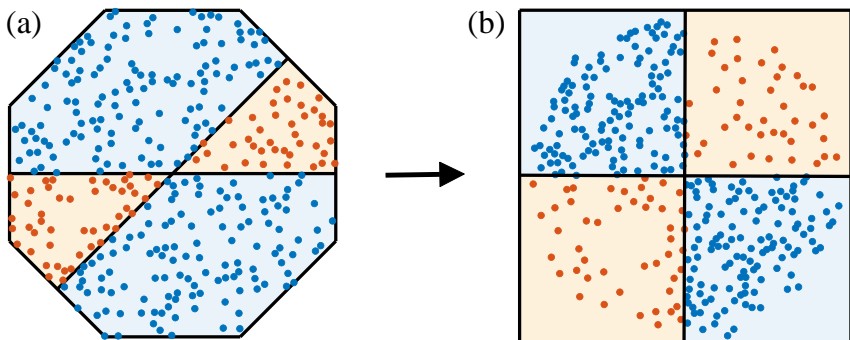

Figure 2: Schematics of how to change site positions in a regular octagon so as to correctly evaluate the quadrupole moment. In a general case, if $m\pi \le \theta \le m\pi + 2\pi/p$ ($m = 0$ or $1$) where $\theta$ is the polar angle of a site inside a regular $p$-gon, then we change the angle to $\theta_d = m\pi + p(\theta - m\pi)/4$, and if $m\pi + 2\pi/p \le \theta \le (m+1)\pi$, then we change it to $\theta_d = [p\theta + (p-4)(m+1)\pi]/[2(p-2)]$. In the figure, we consider the case with $p = 8$. In this case, the sites in the light orange (blue) region in (a) are moved to the first or third (second or fourth) quadrant in (b) without changing the distances from the center.

so that $\overline{R}_j = \{(D_{C_p})^n \mathcal{S}_j : n = 0, 1, \dots, p-1\}$. Clearly, $\overline{R}_j$ respects the $p$-fold rotational symmetry [see Fig. 1(d) for $\overline{R}_1$ when $p = 8$]. We note that although such an operation may break the constraint on the minimal interatomic distance so that sub-gap states appear, their existence does not affect the calculation of topological invariants (see App. C for detailed discussion). We then construct the Hamiltonian $H_j(k_1, \dots, k_{p/2})$ with $k_i \in [0, 2\pi]$ for $i = 1, \dots, p/2$ under twisted boundary conditions on the lattice $\overline{R}_j$ by introducing the momentum $k_1, \dots, k_{p/2}$ (see App. A for details on how to construct the momentum space Hamiltonian). Due to the restored symmetry for the lattice, we obtain a Hamiltonian $H_j(k_1, \dots, k_{p/2})$ that respects the $C_p M$ symmetry, i.e., $U_{C_p M} H_j(k_1, \dots, k_{p/2})(U_{C_p M})^{-1} = H_j(-k_{p/2}, k_1, \dots, k_{p/2-1})$, where $U_{C_p M}$ is a matrix representation of the symmetry in momentum space.

We now define a topological invariant based on the Hamiltonian $H_j$ at the high-symmetry momenta $\boldsymbol{k}_0 = (0, \dots, 0)$ and $\boldsymbol{k}_\pi = (\pi, \dots, \pi)$ where $U_{C_p M}$ and $H_j(\boldsymbol{k}_\mu)$ ($\mu = 0$ or $\pi$) commute [30,31]. Since $(U_{C_p M})^p = -1$, the eigenvalues of $U_{C_p M}$ take the form of $\omega_n = e^{i\pi n/p}$ with $n = \pm 1, \pm 3 \dots, \pm(p-1)$ associated with eigenvectors $|\omega_{n,m}\rangle$ with $m = 1, \dots, 4N_s$, where $N_s$ is the number of sites in $\mathcal{S}_j$. Because of the symmetry $\Xi_2 = \tau_y \sigma_y \kappa$, i.e., $\Xi_2 H_j(\boldsymbol{k}) \Xi_2^{-1} = -H_j(-\boldsymbol{k})$, $|\omega_{n,m}\rangle$ and $|\omega_{-n,m}\rangle$ are connected by $\Xi_2$. We then restrict $H_j$ at high-symmetry momenta to the union of two eigenspaces of $U_{C_p M}$ with eigenvalue $\omega_{\pm n}$, which is labeled as $H_{j,n}$ with $n = 1, 3, \dots, (p-1)$. The restricted Hamiltonian $H_{j,n}(\boldsymbol{k}_\mu)$ belongs to the D class in zero dimension (see App. B). This Hamiltonian can be transformed into an antisymmetric matrix, and thus one can define a $\mathbb{Z}_2$ invariant $v_{j,n}(\boldsymbol{k}_\mu)$ as the sign of the Pfaffian of the antisymmetric matrices. Given that $v_{j,n}(\boldsymbol{k}_0) = v_{j,n}(\boldsymbol{k}_\pi)$ in the atomic limit, one can further define a $\mathbb{Z}_2$ invariant $\chi_{j,n}$ as

$$\chi_{j,n} = (1 - v_{j,n})/2, \tag{3}$$

where $v_{j,n} = v_{j,n}(\boldsymbol{k}_0)v_{j,n}(\boldsymbol{k}_\pi)$. We numerically show that in the thermodynamic limit, $\chi_{j,n}$ for distinct $n$ becomes equal when $p = 8$ (see App. B). We thus identify $H_j$ as topologically nontrivial when at least one of $\chi_{j,n}$ is equal to 1 (denoted as $\chi_j = 1$) and as topologically trivial otherwise (denoted as $\chi_j = 0$). For a single sample configuration, we therefore define

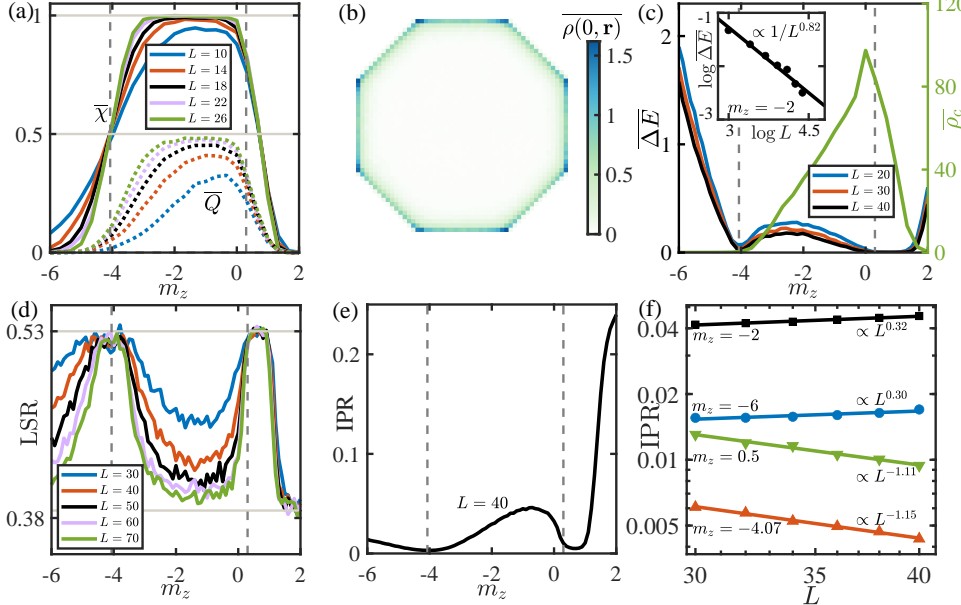

Figure 3: (a) The configuration averaged topological invariant $\overline{\chi}$ (solid lines) and the quadrupole moment $\overline{Q}$ (dotted lines) versus $m_z$ for different system sizes, which are calculated using reconstructed lattice structures $\{\overline{R}_j\}$. (b) The configuration averaged local DOS $\overline{\rho(E, \boldsymbol{r})}$ at zero energy for $m_z = -1$ and $L = 50$, illustrating the existence of corner modes. (c) The configuration averaged bulk energy gaps $\overline{\Delta E}$ calculated under periodic boundary conditions (left vertical axis) and the corner DOS at zero energy $\overline{\rho_c}$ (right vertical axis) versus $m_z$. The green line denotes $\overline{\rho_c}$ for $L = 26$, and the other colored lines denote $\overline{\Delta E}$ for distinct system sizes. Inset: the average gap $\overline{\Delta E}$ versus $L$ for $m_z = -2$, showing a power-law decay as $\overline{\Delta E} \propto 1/L^{\alpha}$ with $\alpha = 0.82$. (d) The configuration averaged LSR versus $m_z$ for different system sizes. (e) The configuration averaged IPR of eigenstates near zero energy as a function of $m_z$ for systems with size $L = 40$. (f) The log-log plot of the IPR with respect to the system size $L$ for different $m_z$. In (a), (c), (d) and (e), the higher-order topological phase is separated from other phases by the vertical dashed grey lines. The number of random configurations in (a), (c) and (e) is 200, in (d) and (f) is 400, and the one in (b) is 2000. Here, $g = 1$.

its topological invariant as an average of $\chi_j$, that is,

$$\chi = \frac{1}{p} \sum_{j=1,\dots,p} \chi_j. \tag{4}$$

As the system also respects chiral symmetry, it raises the question of whether the quadrupole moment [60, 61] can serve as a method of topology characterization of the new topological phase. This is due to the established notion that chiral symmetry plays a role in preserving the quantization of the quadrupole moment [62, 63]. The quadrupole moment was originally proposed to characterize the higher-order topological phase in a geometry with square boundaries harboring four corner modes defined by [60–63]

$$Q = \left[ \frac{1}{2\pi} \text{Im} \log \det(U_o^{\dagger} \hat{D} U_o) - Q_0 \right] \mod 1, \tag{5}$$

where $U_o = \left( |\psi_1\rangle, |\psi_2\rangle, \cdots, |\psi_{n_c}\rangle \right)$, $|\psi_j\rangle$ is the $j$th occupied eigenstate of $H$ with $j = 1, \dots, n_c$ and $n_c = 2N$, and $\hat{D} = \text{diag} \left\{ e^{2\pi i x_j y_j / L^2} \right\}_{j=1}^{4N}$, where $(x_j, y_j)$ is the real space coordinate of

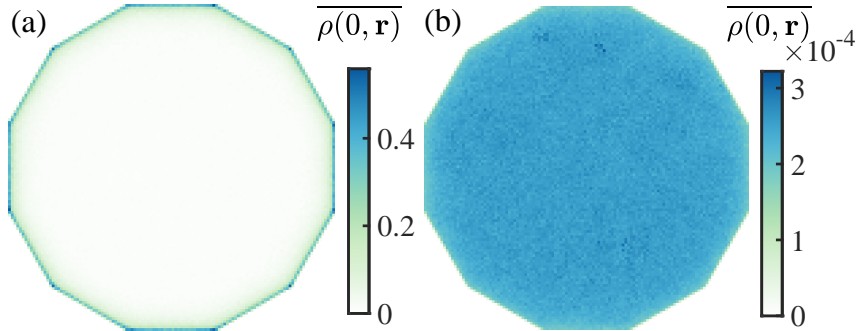

Figure 4: The local DOS $\overline{\rho(E,\mathbf{r})}$ at zero energy averaged over 4000 random configurations in amorphous systems with $p = 12$ for (a) $m_z = -0.5$ and (b) $m_z = -5$ corresponding to a topologically nontrivial and trivial phases, respectively. Here, $g = 1$ and the system size $L = 60$.

the $j$th degree of freedom. Here, $Q_0$ arises from the background positive charge distribution. For a topologically nontrivial phase, the quadrupole moment is quantized to the value of 0.5. However, in our case with more than four corner modes, directly applying this formula will give a zero quadrupole moment because there are even number of corner modes in each quadrant in a Cartesian coordinate system.

To build a reliable quadrupole moment as a topological invariant using the eigenstates of our Hamiltonian $H$, we propose the following method. We change site positions from $(r, \theta)$ to $(r, \theta_d)$ in a polar system so that the sites in a pair of $1/p$ sectors are moved into the first and third quadrants, while the sites in other sectors are moved into the second and fourth quadrants as shown in Fig. 2. Due to the change of site positions, $\hat{D}$ and $Q_0$ should be changed accordingly. We then apply Eq. (5) to calculate the quadrupole moment using the $U_0$ for the Hamiltonian $H$. In practice, for a better result for a finite-size system, we calculate the quadrupole moment for the reconstructed lattice $\{\overline{R}_j\}$ and take their average value as the topological invariant for a sample configuration.

Recently, we have also applied the generalized topological invariant to characterize the higher-order topology of quasicrystalline semimetals in 3Ds [67]. In addition, we generalized the quadrupole moment to characterize the chiral and helical hinge modes in 3D quasicrystals [67].

Figure 3(a) illustrates our numerically calculated topological invariant $\overline{\chi}$ and quadrupole moment $\overline{Q}$ averaged over 200 random configurations. We see that when $-4.07 < m_z < 0.3$, $\overline{\chi}$ and $\overline{Q}$ approach the quantized value of 1 and 0.5, respectively, as we increase the system size, indicating the existence of topologically nontrivial phase. The transition point at $m_z \approx -4.07$ is identified as the crossing point of $\overline{\chi}$ for different system sizes. Further calculation of the zero-energy local DOS shows the existence of eight corner modes in the nontrivial phase, implying that the phase is a second-order topological one [see Fig. 3(b)]. Such eight corner modes are protected by the average $C_8 M$ symmetry. Note that the local DOS is defined as $\rho(E, \mathbf{r}) = \sum_{i,j} \delta(E - E_i) |\Psi_{i,j}(\mathbf{r})|^2$, where $\Psi_{i,j}(\mathbf{r})$ is the $j$th component of the $i$th eigenstate at site $\mathbf{r}$ corresponding to the eigenenergy $E_i$ calculated under open boundary conditions. The topological phase transition with respect to $m_z$ is also evidenced by the bulk energy gap closing as shown in Fig. 3(c). When the system evolves into a trivial phase, the local DOS at a corner vanishes [see Fig. 3(c)]. Although in the topological phase the bulk energy gap is finite for a finite-size system, the gap exhibits a power-law decay [see the inset in Fig. 3(c)], implying that in the thermodynamic limit the system is gapless in the topological phase. Note that near $m_z \approx 0.3$, $\overline{\chi}$ lies between 0 and 1 and does not converge to 0 or 1 as $L$ increases, implying the existence of an intermediate region with the coexistence of topologically nontrivial and trivial

samples. In this region, the gapless bulk states are extended as shown in Fig. 3(d).

To identify the localization property of the topological phase, we further calculate the level-spacing ratio (LSR) and the inverse participation ratio (IPR). The LSR is defined as

$$r_{\text{LSR}} = \frac{1}{N_E - 2} \sum_i \min(\delta_i, \delta_{i+1}) / \max(\delta_i, \delta_{i+1}), \tag{6}$$

where $\delta_i = E_i - E_{i-1}$ with $E_i$ being the $i$th eigenenergy (calculated under periodic boundary conditions), which is sorted in an ascending order, and $\sum_i$ denotes the sum so that $N_E$ energy levels with positive energy closest to the zero energy are counted. To investigate the localization property, one can also calculate the real-space IPR at the energy $E$ defined as

$$I(E) = \frac{1}{N_E} \sum_i \sum_{\boldsymbol{r}} (\sum_{j=1}^{4} |\Psi_{i,j}(\boldsymbol{r})|^2)^2, \tag{7}$$

which gives another quantitative measure of whether states near the energy $E$ are spatially localized.

Figure 3(d) shows that in the topologically nontrivial regime, as we increase the system size, the LSR approaches 0.386 corresponding to the Poisson statistics [68], indicating that the states near zero energy are localized. The states in the vicinity of the transition point exhibit delocalized properties as the LSR sharply rises to the value near 0.53 corresponding to the Gaussian orthogonal ensemble [68] (the Hamiltonian $H$ can be transformed into a real symmetric matrix by a unitary transformation $U_r = e^{-i\frac{\pi}{4}\tau_z\sigma_x}$). In addition, we plot the sample averaged IPR with respect to $m_z$ in Fig. 3(e). We see clearly that the IPR is small at the transition point and the intermediate critical region and is large in the Anderson localized regions, which is consistent with the results of the LSR. We also plot the scaling of the IPR with system sizes in Fig. 3(f), illustrating that at the transition point and the intermediate critical region, the IPR exhibits a power law decrease with system sizes. This further indicates that the bulk states near zero energy are delocalized in these regions. However, in other regions, the IPR exhibits a slight increase with system sizes, suggesting that the states are localized. Note that the increase behavior arises from finite-size effects [62]. In other words, the positive slope declines with system sizes so that it approaches zero in the thermodynamic limit.

We now proceed to study the model in Eq. (1) for $p = 12$ with a regular dodecagon as boundaries. In this case, the system respects $\Xi_2$ symmetry, chiral symmetry and the average $C_{12}M$ and $C_{12}T$ symmetry. In Fig. 4, we plot the configuration averaged local DOS at zero energy $\overline{\rho(0, \boldsymbol{r})}$, showing the presence and absence of twelve corner modes in the topologically nontrivial and trivial phases, respectively. Owing to the average crystalline symmetry, the local DOS respects the $C_{12}$ symmetry. Although we demonstrate our theory using the typical cases with $p = 8$ or $p = 12$, it is applicable for other average rotational symmetries in amorphous systems.

## 4 Conclusion

In summary, we have shown that amorphous systems can allow for the existence of new higher-order topological insulators (gapless but localized) hosting eight or twelve zero-energy corner modes, which cannot exist in crystalline systems. Such topological phases are protected by a crystalline symmetry on average. We also propose two topological invariants to characterize the topological phases. Higher-order topological phases in crystalline lattices have been experimentally observed in phononic [69], microwave [70], electric circuit [71], and photonic systems [72], and we thus expect that the new topological phases in amorphous lattices may

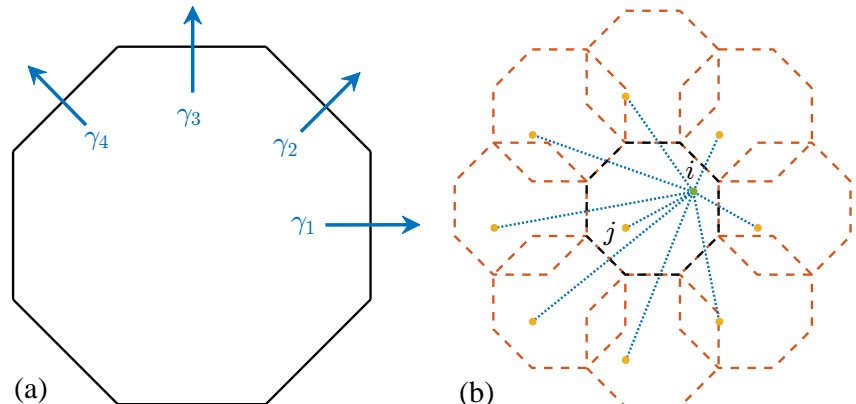

Figure 5: (b) Resulted polygon configurations by applying translation operators $\gamma_1$, ..., $\gamma_4$, $\gamma_1^{-1}$, ..., and $\gamma_4^{-1}$ to a regular octagon in (a). For clarity, we also plot the translated sites from the site $j$ as solid yellow circles. The dotted lines describe the hopping from the site $i$ to the site $j$ under TBCs.

be realized in these systems given their high controllability and versatility. In addition, given that a very recent experiment shows the existence of topological amorphous $Bi_2Se_3$ [24], our work may also inspire the interest of exploring the new topological phases without crystalline counterparts in amorphous solid-state materials.

While ideal amorphous materials are typically isotropic, anisotropy can be introduced in amorphous materials by applying magnetic fields [73] or through interface interactions [74]. Doping magnetic impurities could enable the realization of the higher-order topological phase in amorphous materials. Ideally, magnetic impurities with spin up can be doped in one $1/p$ sector, while spin-down impurities can be doped in the adjacent sectors. Similarly, the remaining sectors should be doped following the same pattern, resulting in alternate magnetic moments of up and down in the $1/p$ sectors. This approach breaks the average continuous rotational symmetry of amorphous materials, while still maintaining the average $C_pT$ symmetry. Another approach is to apply local magnetic fields under the substrate with alternating directions for each $1/p$ sector during the growing process of amorphous materials. An alternative could be to realize our model as either a highly disordered quasicrystal or an amorphous layer on a quasicrystal.

## Acknowledgments

We thank K. Li and X. Li for helpful discussions.

**Funding information** The work is supported by the National Natural Science Foundation of China (Grant No. 11974201), Tsinghua University Dushi Program and Innovation Program for Quantum Science and Technology (Grant No. 2021ZD0301604).

## A  Momentum-space Hamiltonian

In this section, we will construct the momentum-space Hamiltonian by using the twisted boundary conditions (TBCs) for a system whose sites are distributed in a regular $p$-gon.

In a geometry of a regular $p$-gon with the width of $L$, we define translation operators $\gamma_\mu$ with $\mu = 1, \ldots, p/2$ so that applying it to a site position vector $\boldsymbol{r}$ results in $\gamma_\mu(\boldsymbol{r}) = \boldsymbol{r} + \boldsymbol{T}_\mu$ with $\boldsymbol{T}_\mu = L(\cos\theta_\mu \boldsymbol{e}_x + \sin\theta_\mu \boldsymbol{e}_y)$ and $\theta_\mu = 2\pi(\mu - 1)/p$. Figure 5 displays the resulted positions by applying $\gamma_1, \ldots, \gamma_4, \gamma_1^{-1}, \ldots,$ and $\gamma_4^{-1}$, respectively, to the position vector of the site $j$ in the case with $p = 8$. For the twisted Hamiltonian, besides the hopping between the sites inside the polygon, we also need to consider the hopping between the sites inside the polygon and the sites outside the polygon resulted from the translation operations. Different from the case with square boundaries, one cannot use regular $p$-gons for $p > 6$ to tessellate the entire 2D Euclidean plane. For the latter hopping, we thus only consider the sites outside the polygon obtained by applying each translation operator in $\{\gamma_1, \ldots, \gamma_{p/2}, \gamma_1^{-1}, \ldots, \gamma_{p/2}^{-1}\}$ to the internal sites. For example, for a site $\boldsymbol{r}_j$, after these operations, we obtain a set of sites,

$$\{\boldsymbol{r}_j(n_\mu) = \boldsymbol{r}_j + n_\mu \boldsymbol{T}_\mu : \mu = 1, \ldots, p/2 \text{ and } n_\mu = -1, 0, 1\}, \tag{A.1}$$

where we also include the original site $\boldsymbol{r}_j$.

For the hopping from the site $i$ to the site $j$ in the twisted Hamiltonian, we need to involve the hopping from the site $i$ to all the sites in $\{\boldsymbol{r}_j(n_\mu)\}$ [see Fig. 5(b)] by imposing an extra phase $e^{-in_\mu k_\mu}$ so that the hopping matrix is expressed as

$$T_{\boldsymbol{r}_j - \boldsymbol{r}_i}(k_1, \ldots, k_{p/2}) = \sum_{\{\boldsymbol{r}_j(n_\mu)\}} T[\boldsymbol{r}_j(n_\mu) - \boldsymbol{r}_i] e^{-in_\mu k_\mu}. \tag{A.2}$$

As a result, the twisted Hamlitonian is given by

$$\hat{H}(k_1, \ldots, k_{p/2}) = \sum_{\boldsymbol{r}} \left[ m_z \hat{c}_{\boldsymbol{r}}^\dagger \tau_z \sigma_0 \hat{c}_{\boldsymbol{r}} + \sum_{\boldsymbol{d}} \hat{c}_{\boldsymbol{r}+\boldsymbol{d}}^\dagger T_{\boldsymbol{d}}(k_1, \ldots, k_{p/2}) \hat{c}_{\boldsymbol{r}} \right], \tag{A.3}$$

where $k_j \in [0, 2\pi]$ with $j = 1, \ldots, p/2$. To achieve periodic boundary conditions, we take $k_j = 0$ with $j = 1, \ldots, p/2$.

# B  Derivation of a $\mathbb{Z}_2$ topological invariant

In this section, we will provide a detailed derivation of a $\mathbb{Z}_2$ topological invariant based on the $C_p M$ symmetry (see also Ref. [30] for the quasicrystal case).

Since $(U_{C_p M})^p = -1$, the eigenvalues of $U_{C_p M}$ take the form of $\omega_n = e^{i\pi n/p}$ with $n = \pm 1, \pm 3, \ldots, \pm(p-1)$ associated with eigenvectors $|\omega_{n,m}\rangle$ with $m = 1, \ldots, 4N_s$, where $N_s$ is the number of sites in $S_j$. Due to the symmetry $\Xi_2 = \tau_y \sigma_y \kappa$, $\Xi_2 |\omega_{n,m}\rangle$ is an eigenvector of $U_{C_p M}$ with an eigenvalue $\omega_{-n}$, and $\Xi_2$ thus connects the eigenspaces of $\omega_{\pm n}$.

We now restrict the Hamiltonian $H_j(\boldsymbol{k}_\mu)$ ($\mu = 0$ or $\pi$) to a subspace spanned by $|\omega_{n,m}\rangle$ and $\Xi_2 |\omega_{n,m}\rangle$ with $m = 1, \ldots, 4N_s$ for a certain $n$, that is,

$$H_{j,n}(\boldsymbol{k}_\mu) = U_n^\dagger H_j(\boldsymbol{k}_\mu) U_n, \tag{B.1}$$

where $U_n$ is a $4pN_s \times 8N_s$ matrix defined as

$$U_n = (|\omega_{n,1}\rangle, \ldots, |\omega_{n,4N_s}\rangle, \Xi_2 |\omega_{n,1}\rangle, \ldots, \Xi_2 |\omega_{n,4N_s}\rangle). \tag{B.2}$$

Since $H_j(\boldsymbol{k}_\mu)$ commutes with $U_{C_p M}$, we can reduce the restricted Hamiltonian to the following form

$$H_{j,n}(\boldsymbol{k}_\mu) = \begin{pmatrix} V_n^\dagger H_j(\boldsymbol{k}_\mu) V_n & 0 \\ 0 & (\Xi_2 V_n)^\dagger H_j(\boldsymbol{k}_\mu) \Xi_2 V_n \end{pmatrix} = \begin{pmatrix} V_n^\dagger H_j(\boldsymbol{k}_\mu) V_n & 0 \\ 0 & -\left[ V_n^\dagger H_j(\boldsymbol{k}_\mu) V_n \right]^* \end{pmatrix}, \tag{B.3}$$

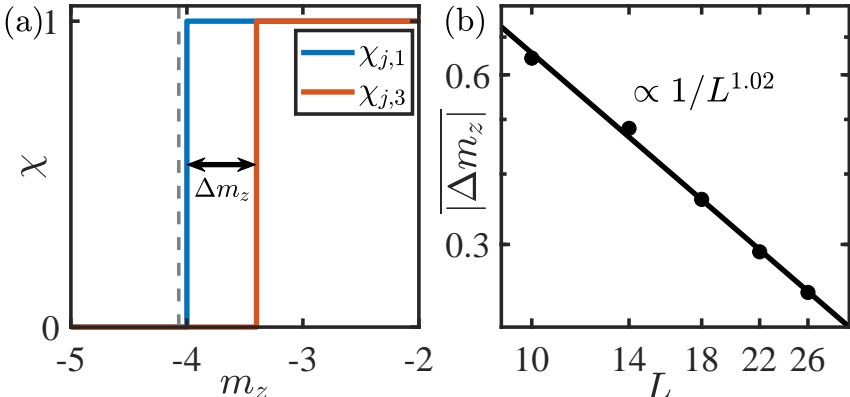

Figure 6: (a) The calculated $\chi_{j,1}$ and $\chi_{j,3}$ with respect to $m_z$ for a typical random configuration. (b) The configuration averaged difference $\overline{|\Delta m_z|}$ between transition points of $\chi_{j,1}$ and $\chi_{j,3}$ versus $L$ around $m_z \approx -4$. The number of random configurations is 200. Here, $g = 1$.

where $V_n = (|\omega_{n,1}\rangle, \ldots, |\omega_{n,4N_s}\rangle)$ is a $4pN_s \times 4N_s$ matrix. $H_{j,n}(\boldsymbol{k}_\mu)$ thus respects an antiunitary antisymmetry $X = U_X \kappa$ where $U_X = s_x \otimes I_{4N_s}$ and $I_{4N_s}$ is a $4N_s \times 4N_s$ identity matrix, i.e., $X H_{j,n}(\boldsymbol{k}_\mu) X^{-1} = -H_{j,n}(\boldsymbol{k}_\mu)$ so that it belongs to the D class in zero dimension. Its topology can be characterized by the sign of the Pfaffian of an antisymmetric matrix $H_{j,n}^{\mathrm{a}}(\boldsymbol{k}_\mu)$ transformed from $H_{j,n}(\boldsymbol{k}_\mu)$. To obtain the antisymmetric Hamiltonian, we first use the Autonne-Takagi factorization [75,76] to write the symmetric unitary matrix $U_X$ as $U_X = V_X D V_X^T$ where $D = \mathrm{diag}\{e^{\mathrm{i}\varphi_j}\}_{j=1}^{8N_s}$ is a diagonal matrix consisting of the eigenvalues of $U_X$, and $V_X = (|\varphi_1\rangle, \ldots, |\varphi_{8N_s}\rangle)$ with $|\varphi_j\rangle$ being the eigenvector with eigenvalue $e^{\mathrm{i}\varphi_j}$ for $j = 1, \ldots, 8N_s$. We now apply the matrix $W = \sqrt{D^*}V_X^\dagger$ so as to obtain $WXW^\dagger = \kappa$ and $H_{j,n}^{\mathrm{a}}(\boldsymbol{k}_\mu) = W H_{j,n}(\boldsymbol{k}_\mu) W^\dagger$. It follows that $[H_{n,j}^{\mathrm{a}}(\boldsymbol{k}_\mu)]^* = [H_{n,j}^{\mathrm{a}}(\boldsymbol{k}_\mu)]^T = -H_{n,j}^{\mathrm{a}}(\boldsymbol{k}_\mu)$.

We thus can define a $\mathbb{Z}_2$ invariant $\nu_{j,n}(\boldsymbol{k}_\mu)$ as the sign of the Pfaffian of the antisymmetric matrix $H_{n,j}^{\mathrm{a}}(\boldsymbol{k}_\mu)$. Given that $\nu_{j,n}(\boldsymbol{k}_0) = \nu_{j,n}(\boldsymbol{k}_\pi)$ in the atomic limit, one can further define a $\mathbb{Z}_2$ invariant $\chi_{j,n}$ as

$$\chi_{j,n} = (1 - \nu_{j,n})/2, \tag{B.4}$$

where $\nu_{j,n} = \nu_{j,n}(\boldsymbol{k}_0)\nu_{j,n}(\boldsymbol{k}_\pi)$. Similarly, one can also evaluate the invariant in the eigenspaces of the symmetry $(C_p M)^2$. In this case, $|\omega_{n,m}\rangle$ and $|\omega_{n-p,m}\rangle$ share the same eigenvalue for the operator $(C_p M)^2$ so that the $\mathbb{Z}_2$ invariant is given by $(1 - \nu_{j,n}\nu_{j,p-n})/2$. Since the topology is protected by the $C_p M$ symmetry instead of the $(C_p M)^2$ symmetry, this invariant should vanish, leading to $\nu_{j,n} = \nu_{j,p-n}$ and $\chi_{j,n} = \chi_{j,p-n}$ with $n = 1, 3, \ldots, p/2 - 1$. In the case with $p = 8$, we have $\chi_{j,1} = \chi_{j,7}$ and $\chi_{j,3} = \chi_{j,5}$. In a single sample, $\chi_{j,1}$ is not equal to $\chi_{j,3}$ near the phase transition point as shown in Fig. 6(a). However, numerical results show that they give the same transition point in the thermodynamic limit as revealed by Fig. 6 showing the difference $\overline{|\Delta m_z|}$ between the transition points determined by $\chi_{j,1}$ and $\chi_{j,3}$. It displays a power-law decay as $\overline{|\Delta m_z|} \propto 1/L^{1.02}$, indicating that $\chi_{j,1}$ is equal to $\chi_{j,3}$ in the thermodynamic limit.

We thus identify $H_j$ as topologically nontrivial when one of $\chi_{j,n}$ is equal to 1 (denoted as $\chi_j$). For a single sample configuration, we therefore define its topological invariant as an average of $\chi_j$, that is,

$$\chi = \frac{1}{p}\sum_{j=1,\ldots,p} \chi_j. \tag{B.5}$$

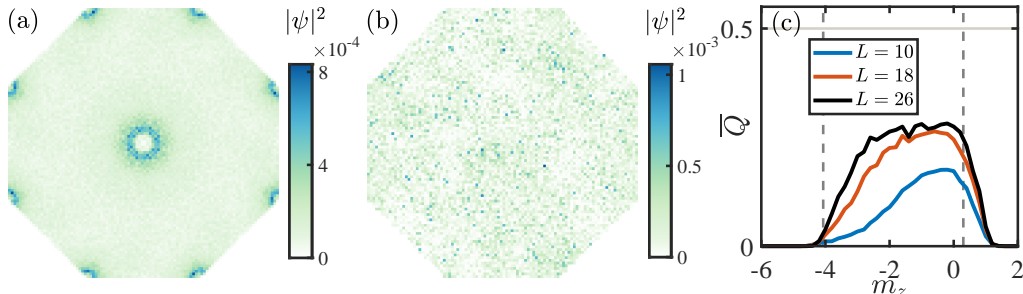

Figure 7: The configuration averaged density distribution of two low-energy states near zero energy for (a) symmetry restored systems and (b) original systems without rotational symmetry under periodic boundary conditions. Here, $m_z = -1$. (c) The sample averaged quadrupole moment versus $m_z$ for original systems with distinct system sizes. Here, $g = 1$ and the number of random configurations is 200 in (a) and (b) and is 1600 in (c).

## C    Sub-gap states in symmetry restored systems

In the main text, we restore the rotational symmetry in order to define the $\mathbb{Z}_2$ topological invariant $\chi$. Such an operation may break the constraint on the minimal interatomic distance. As a result, the sub-gap states arise in the symmetry restored systems as shown in Fig. 7(a), where the configuration averaged density distribution of two low-energy states near zero energy under periodic boundary conditions is plotted. However, we would like to clarify the following two points:

1. We only use the symmetry restored system to calculate the topological invariants shown in Fig. 3(a). For all the other results, we consider random lattices without restoring the symmetry. For example, in Fig. 3(c), we calculate the energy gap for random lattices without restoring the symmetry so that no sub-gap physics arises from the gluing operations. This can be clearly seen in Fig. 7(b) where the density distribution does not exhibit any sub-gap states at the glueing interfaces. Thus, the gapless behavior arises from the bulk states.

2. To calculate the topological invariant in Fig. 3(a), we consider restoring the rotational symmetry. Thus, the topological invariant $\chi$ can be constructed. For the quadrupole moment, the rotational symmetry is not required. One can directly calculate the quadrupole moment for the original system without restoring the symmetry. We find that the quadrupole moment exhibits nonzero values in the topological region and its value increases toward 0.5 with system size [see Fig. 7(c)], similar to the quadrupole moment calculated for the symmetry restored system in Fig. 3(a). In addition, the calculated topological phase transition points agree well with the results of the energy gap and localization properties. All these results suggest that despite the presence of introduced sub-gap states, their existence does not affect the calculation of topological invariants.

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
