# Peer review of "Average Symmetry Protected Higher-order Topological Amorphous Insulators"

_SciPost Physics, doi:SciPost Phys. 15, 193 (2023)_

## Round 1 · Referee Report · Anonymous (Referee 1) · 2023-7-9

Strengths

The paper shows that higher order phases which are protected by rotational symmetries - which cannot be achieved in crystals - and were shown to be realised in quasi-crystals can also be realised in amorphous systems.

1) To that effect the authors show that while the boundary modes get realised in any typical system; to well-define a topological invariant they need to make some equivalent systems where the lattice is deliberately made rotationally invariant and a band structure can be made by artificially repeating the pattern. They show that the region where the Pfaffian of this system is non trivial - is consistent with non-trivial phase of the system and associated gap closings. 2) They also calculate the real space quadrupolar moment in the system by deforming the lattice into a square patch which is consistent with these transition values. 3) To further investigate the low energy modes near half-filling they look at typical level spacing ratio and find that transitions between topological and trivial phases are characterised by change in ratio from GOE ensemble to poissonian which is interesting.

Weaknesses

No particular weaknesses

Report

I find the paper interesting and recommend publication. I particularly find the momentum space construction and the invariant calculation interesting and it might be useful for the community of researchers working on topological phases away from crystalline settings. I was wondering if the IPR of the states close to Fermi energy show any anomalous scaling near the transitions as a function of $m_z$. It will be useful if the authors can comment on that.
  • validity: high
  • significance: good
  • originality: good
  • clarity: high
  • formatting: good
  • grammar: good

Author:  Yong Xu  on 2023-10-13  [id 4038]

(in reply to Report 1 on 2023-07-09)

Please find our response to the Report in the attachment.

Attachment:

reply_to_referee1.pdf

---

## Round 1 · Referee Report · Daniel Varjas (Referee 2) · 2023-7-17

Strengths

1) Clearly written, good presentation. 2) Discusses the timely problem of amorphous topological phases, and more broadly, topological phases protected by non-crystalline spatial symmetries. 3) Novel use of "momentum-space" invariants derived from twisted periodic boundary conditions with more k components than spatial dimensions.

Weaknesses

1) My main concern is the novelty of the results compared to the earlier works on quasicrystals (refs 28 and 29), and the relevance of the results to physical amorphous materials. The p-fold rotational symmetries (particularly p=8, 10, 12) arise naturally in quasicrystals, but seem artificial in an amorphous system. Amorphous materials are typically isotropic, possessing continuous rotation symmetry on average, lacking any preferred directions. In my view, the systems studied here are better described as highly disordered quasicrystals - or perhaps an amorphous layer on a quasicrystal substrate - inheriting the discrete rotational symmetry in the hopping Hamiltonian, but the atomic positions completely disordered. Do the authors agree with this view? Can the authors suggest other physical setups that these models can describe? I believe that these consideration should be discussed at some point of the manuscript to make the context of the research clear to the reader, and motivate the relevance of the results to physically realizable amorphous materials.

2) What guarantees that there are no gapless modes at the boundary of the segments after restoring the rotational symmetry when constructing the Hamiltonians H_j, or when applying periodic boundary conditions? As described in the text, the amorphous structure is not completely uncorrelated, rather it is a random set of hard disks. Is this constraint on the minimal interatomic distance obeyed by the symmetrized systems? Based on the description of the construction, it is not, which may result in additional sub-gap states at the gluing interfaces in the symmetrized and periodic boundary condition systems. I ask the authors to clarify these questions and demonstrate that no new subgap physics arises at the gluing interfaces, for example by examining the spatial localization of the low-energy states that appear at large system sizes (fig 3c inset).

3) Why does the invariant chi_n not depend on n? The manuscript only presents numerical evidence for chi_1 = chi_3. I would expect a similar scenario as in ref 28 SM, when additional symmetries forcing a vanishing Chern-number are responsible for this equality, and is not forced by protecting symmetries. It would be worth to check whether this is the case, or at least comment on the possibility.

4) Why is it necessary (or advantageous) to distort the system to calculate the quadrupole moment? Is the quantity measured this way really the quadrupole moment, or perhaps some higher moment? It is really unclear to me how a procedure like this, manifestly breaking the protecting rotational symmetries, can extract a topological invariant.

Report

The manuscript studies topological phases of amorphous systems protected by chiral and average p-fold (p even) rotation symmetries. The analysis is carried out using a tight-binding model on a random graph, using "momentum-space" invariants derived from twisted periodic boundary conditions, quadrupole moment, and spectral signatures. The manuscript is well written, and the results are sound.

In my opinion the manuscript is an interesting, but fairly straightforward generalisation of earlier work on quasicrystalline topological phases with added disorder (see point 1 of the Weaknesses section for details). Hence I recommend moving the manuscript to SciPost Physics Core, and publication with minor clarifications; or would request the authors to further support the novelty of the work, and motivate why the paper meets the acceptance criteria of SciPost Physics in light of the questions raised.

Requested changes

Minor change requests below, for major questions see the Weaknesses section.

1) The hopping Hamiltonian in eqn. 1 for p=8 is (up to a basis transformation, and inclusion of bond-length-dependent prefactors) is identical to that of ref [28], and the generalisation of the last term for cases with p other than 8 was presented in ref [29], I ask the authors to make this clear by citations when introducing the Hamiltonian.

2) The following statement is unclear to me: “Note that near mz ≈ 0.3 there appears an intermediate region with the coexistence of topologically nontrivial and trivial samples where the gapless bulk states are extended as shown in Fig. 3(d).” Should clarify the text or fig. 3 to make this clear.

3) Should mention in the caption what the vertical dashed grey lines in fig. 3 denote.

4) "high-symmetric momenta" should be, following standard terminology, "high-symmetry momenta"

  • validity: high
  • significance: good
  • originality: good
  • clarity: high
  • formatting: perfect
  • grammar: excellent

Author:  Yong Xu  on 2023-10-13  [id 4037]

(in reply to Report 2 by Daniel Varjas on 2023-07-17)
Category:
answer to question

Please find our response to the Report in the attachment.

Attachment:

reply_to_referee2.pdf

---

## Round 3 · Referee Report · Daniel Varjas (Referee 2) · 2023-10-18

Strengths
Weaknesses
1) It is still unclear to me why the χ1 = χ3 equality can't be established as a consequence of C=0. While I agree with the authors' response, it would be good to clarify in the manuscript that this equality might only hold in certain phases. 2) Unfortunately the SM of arXiv:2307.14974 is not available, so I could not consult it on further detalis of the distorted quadrupole moment. I still have doubts about the physical meaning of this invariant, but I find the updated manuscript sufficiently clear and convincing about its practical usefulness.
Report
Requested changes
See the "Weaknesses" section for further suggestions for optional clarifications.
Anonymous on 2023-10-20 [id 4047]
Weakness 1: It is still unclear to me why the $\chi_1=\chi_3$ equality can’t be established as a consequence of C=0.While I agree with the authors’ response, it would be good to clarify in the manuscript that this equality might only hold in certain phases.
Reply: We thank the referee for the agreement with our response. In fact, the consequence of $C=0$ is $(-1)^{\sum_j \chi_j}=1$, which cannot enforce the equality of $\chi_1=\chi_3$. We have followed the referee’s suggestion to clarify in the revised manuscript that “We note that one cannot derive the equality from the vanishing Chern number, and this equality might only hold in certain phases of certain models.”. We have also updated the arXiv version, which will become public soon.
Weakness 2: Unfortunately the SM of arXiv:2307.14974 is not available. so I could not consult it on further details of the distorted quadrupole moment. I still have doubts about the physical meaning of this invariant, but I find the updated manuscript sufficiently clear and convincing about its practical usefulness.
Reply: We thank the referee for the very positive comment that “I find the updated manuscript sufficiently clear and convincing about its practical usefulness”. We have followed the referee’s nice suggestion to update arXiv:2307.14974 including the SM so that the referee could find more details there. We understand that the physical interpretation of the distorted quadrupole moment remains an open question, as noted by the referee, and merits further investigations.
Report: The authors have sufficiently addressed my main concerns. See the “Weaknesses” section for further suggestions for optional clarifications.
Reply: We thank the referee for the very positive comment that “The authors have sufficiently addressed my main concerns” and nice suggestions. We have incorporated the suggested changes and believe that our manuscript is now ready for publication in SciPost Physics.

---

## Round 3 · Author Response

Warnings issued while processing user-supplied markup:
- Inconsistency: plain/Markdown and reStructuredText syntaxes are mixed. Markdown will be used.
Add "#coerce:reST" or "#coerce:plain" as the first line of your text to force reStructuredText or no markup.
You may also contact the helpdesk if the formatting is incorrect and you are unable to edit your text.
Dear Editor,
Thank you very much for sending us the reports for our manuscript. We sincerely appreciate both referees' time on reviewing the manuscript and their helpful and constructive suggestions/comments, which have helped us improve our paper significantly.
Referee 1 “find the paper interesting and recommend publication” and “particularly find the momentum space construction and the invariant calculation interesting” and “it might be useful for the community of researchers working on topological phases away from crystalline settings.”. We thank the referee for recommending our paper for publication in SciPost Physics and his/her high evaluation and helpful suggestions, which we have carefully accounted for in the revised version.
Referee Dániel Varjas judged our paper as “clearly written, good presentation”, “the results are sound”, “discusses the timely problem of amorphous topological phases, and more broadly, topological phases protected by non-crystalline spatial symmetries” and “novel use of "momentum-space" invariants derived from twisted periodic boundary conditions with more k components than spatial dimensions.”. The referee asked us to clarify the novelty of the work given the concern regarding “the novelty of the results compared to the earlier works on quasicrystals” and “the relevance of the results to physical amorphous materials”. We sincerely appreciate his concerns and helpful suggestions. In this reply, we have clarified the novelty of our work and discussed the experimental accessibility in real materials. We have also carefully addressed all his questions.
Before responding to the questions and comments raised by the referees, we also want to emphasize the following three important major contributions of this work, which make it clearer why our paper is suitable for publication in SciPost Physics:
1) This work is the first one to explore the question of whether or not higher-order topological phases without crystalline counterparts can exist in 2D amorphous systems. We have established the existence of such an exotic phase by giving a concrete model and studied its physical properties. The topological phases are protected by average rotational symmetry, which is different from those in quasicrystals protected by the exact rotational symmetry in nature.
2) We have for the first time introduced the generalized quadrupole moment. This topological invariant not only can characterize the higher-order topology in amorphous systems, but also that in quasicrystals. Compared with the previously introduced $Z_2$ topological invariant, the generalized quadrupole moment has broader applications in the study of higher-order topological phases in non-crystalline systems, e.g., it can be generalized to 3D to characterize the chiral and helical hinge modes in 3D quasicrystals.
3) While both the higher-order topological phases in amorphous and quasicrystalline lattices can support corner modes lacking crystalline equivalents, they have fundamentally distinct properties. In the amorphous case, the topological phase is gapless with a vanishing bulk energy gap, and the bulk states near zero energy are spatially localized corresponding to an Anderson insulator. However, in the quasicrystal case, the topological phase is gapped with a finite bulk energy gap corresponding to a band insulator.
To sum up, we have clarified the novelty of our work and carefully and thoroughly addressed all the comments/suggestions from the referees. We believe that our work is suitable for publication in SciPost Physics.
A summary of the major revisions and a detailed response have been appended in the following.
Thank you.
Yours sincerely,
Yu-Liang Tao, Jiong-Hao Wang, and Yong Xu
Responses to Referee 1:
We thank the referee for the careful reading of our manuscript and the positive comments that “I find the paper interesting”, “I particularly find the momentum space construction and the invariant calculation interesting” and “it might be useful for the community of researchers working on topological phases away from crystalline settings.” We thank his/her recommendation of publication of our work in SciPost Physics. We also appreciate his/her constructive suggestions/questions which are invaluable in improving our paper. The following contains our detailed response to specific points.
Comment: The paper shows that higher order phases which are protected by rotational symmetries - which cannot be achieved in crystals - and were shown to be realised in quasi-crystals can also be realised in amorphous systems.
1) To that effect the authors show that while the boundary modes get realised in any typical system; to well-define a topological invariant they need to make some equivalent systems where the lattice is deliberately made rotationally invariant and a band structure can be made by artificially repeating the pattern. They show that the region where the Pfaffian of this system is non trivial - is consistent with non-trivial phase of the system and associated gap closings.
2) They also calculate the real space quadrupolar moment in the system by deforming the lattice into a square patch which is consistent with these transition values.
3) To further investigate the low energy modes near half-filling they look at typical level spacing ratio and find that transitions between topological and trivial phases are characterised by change in ratio from GOE ensemble to poissonian which is interesting.
Reply: We thank the referee for the nice summary of our results.
Comment: I find the paper interesting and recommend publication. I particularly find the momentum space construction and the invariant calculation interesting and it might be useful for the community of researchers working on topological phases away from crystalline settings.
Reply: We thank the referee for the positive evaluation of our work and recommendation of our work for publication in SciPost Physics.
Comment 1: I was wondering if the IPR of the states close to Fermi energy show any anomalous scaling near the transitions as a function of m_z. It will be useful if the authors can comment on that.
Reply: We thank the referee for the nice suggestion. We have followed the suggestion to calculate the IPR of the states near zero energy with respect to $m_z$ and plot it in the Fig. 3(e) and (f) in the revised manuscript. We see clearly that the IPR is small at the transition point and the intermediate critical region and is large in the Anderson localized regions, which is consistent with the results of the LSR. We also plot the scaling of the IPR with system sizes and find that at the transition point and the intermediate critical region, the IPR exhibits a power law decrease with system sizes, providing further evidence that the bulk states near zero energy are delocalized in these regions. However, in other regions, the IPR exhibits a slight increase with system sizes, suggesting that the states are localized. Note that the increase behavior arises from finite-size effects. In other words, the positive slope declines with system sizes so that it approaches zero in the thermodynamic limit.
Responses to Referee Dániel Varjas:
We thank the referee for carefully reading our manuscript and judging our paper as “clearly written, good presentation”, “the results are sound”, “discusses the timely problem of amorphous topological phases, and more broadly, topological phases protected by non-crystalline spatial symmetries” and “novel use of "momentum-space" invariants derived from twisted periodic boundary conditions with more k components than spatial dimensions.”. The referee's main concerns are “the novelty of the results compared to the earlier works on quasicrystals” and “the relevance of the results to physical amorphous materials”. In the following response, we have addressed these concerns very carefully and made changes as requested.
Comment: 1) Clearly written, good presentation. 2) Discusses the timely problem of amorphous topological phases, and more broadly, topological phases protected by non-crystalline spatial symmetries. 3) Novel use of " momentum-space" invariants derived from twisted periodic boundary conditions with more k components than spatial dimensions.
Reply: We thank the referee for the positive comments of our work.
Comment 1: My main concern is the novelty of the results compared to the earlier works on quasicrystals (refs 28 and 29), and the relevance of the results to physical amorphous materials. The p-fold rotational symmetries (particularly p=8, 10, 12) arise naturally in quasicrystals, but seem artificial in an amorphous system. Amorphous materials are typically isotropic, possessing continuous rotation symmetry on average, lacking any preferred directions. ln my view, the systems studied here are better described as highly disordered quasicrystals - or perhaps an amorphous layer on a quasicrystal substrate - inheriting the discrete rotational symmetry in the hopping Hamiltonian, but the atomic positions completely disordered. Do the authors agree with this view? Can the authors suggest other physical setups that these models can describe? I believe that these considerations should be discussed at some point of the manuscript to make the context of the research clear to the reader, and motivate the relevance of the results to physically realizable amorphous materials.
Reply: We thank the referee for the insightful comment. Below are our detailed responses to the referee’s concerns.
For the novelty of the results compared to the earlier works on quasicrystals (Refs. [30,31], previous Refs. [28,29]), we acknowledge that we drew inspiration from quasicrystals, especially the excellent work Ref. [30] in the definition of the $Z_2$ topological invariant. Nevertheless, our work is far from a straightforward generalization of the quasicrystal case in both the nature of amorphous matter and the construction of two topological invariants.
Previous works show the existence of higher-order topological phase supporting eight corner modes in quasicrystals. The phase is protected by the eight-fold rotational symmetry. A $Z_2$ topological invariant is defined based on the $C_8 M$ symmetry. In addition, prior to our paper, it was still unclear whether the quadrupole moment could be extended to characterize the higher-order topology in quasicrystals without crystalline counterparts. As the referee must know, amorphous systems are short of the exact rotational symmetry protecting the higher-order phases in quasicrystals so that one cannot define the $Z_2$ topological invariant. Therefore, it is not obvious to deduce whether higher-order topological phases can exist in amorphous systems from the previous results in quasicrystals. In fact, the higher-order topological phases in amorphous systems are protected by the average rotational symmetry, different from the exact symmetry in quasicrystals.
Furthermore, to the best of our knowledge, whether all 4n-fold (n is a positive integer) rotational symmetries can arise in quasicrystals remains unclear. For instance, there has been no quasicrystals constructed with sixteen-fold or twenty-fold rotational symmetry [Generalized Dynamics of Soft-Matter Quasicrystals: Mathematical Models, Solutions and Applications (Springer, Singapore, 2022), 2nd ed]. In comparison, amorphous materials can possess any 4n-fold average rotational symmetry, and we can explore richer higher-order topological phases with arbitrary 4n corner modes which cannot be realized in quasicrystals.
For the $Z_2$ topological invariant, in quasicrystals, the effective Hamiltonian is constructed based on the plane wave in the 4D Brillouin zone. However, amorphous lattices cannot be generated from a higher-dimensional space with translational symmetry by a cut-and-project method, thus the construction method of the effective Hamiltonian for quasicrystals are no longer applicable. To fix the problem, we introduce unconventional twisted boundary conditions to construct the effective Hamiltonian in k space. This helps us build the $Z_2$ topological invariant.
Another important contribution of our work is the introduction of how to calculate the quadrupole moment in topological phases without crystalline counterparts in amorphous systems. In fact, such a topological invariant is also applicable to quasicrystals (please see arXiv:2307.14974). Compared with the $Z_2$ index introduced in Ref. [30], the generalized quadrupole moment can also be extended to characterize the chiral and helical hinge modes in 3D quasicrystals (arXiv:2307.14974). For example, one can use the winding number of the generalized quadrupole moment to characterize the chiral mode of 3D second-order topological insulators in quasicrystals without crystalline counterparts. We therefore believe that the generalized quadrupole moment introduced here will play an important role in the study of higher-order topology in non-crystalline systems.
Finally, we would like to emphasize that while both the higher-order topological phases in amorphous and quasicrystal lattices can support corner modes lacking crystalline equivalents, they have fundamentally distinct properties. In the amorphous case, the topological phase is gapless with a vanishing bulk energy gap, and the bulk states near zero energy are spatially localized corresponding to an Anderson insulator. However, in the quasicrystal case, the topological phase is gapped with a finite bulk energy gap corresponding to a band insulator.
Because of the above points, we believe that our work is novel and meets the acceptance criteria of SciPost Physics.
For the relevance of the results to physical amorphous materials, we agree with the referee that ideal amorphous materials are typically isotropic. However, in experiments, anisotropy can be introduced in the growing process by applying magnetic field [J. Appl. Phys. 106, 023918 (2009)] or through interface interactions [Phys. Rev. Lett. 100, 117201 (2008)]. As the referee suggests, using quasicrystals as the substrate is a good way to realize the $C_p T$ symmetry in our model. For general p=4n without quasicrystal counterparts, the $C_p T$ symmetry might be realized by doping magnetic impurities. Ideally, one can dope magnetic impurities with spin up in one 1/p sector and spin-down impurities in its adjacent sectors. Other sectors are doped in the similar manner such that the 1/p sectors have magnetic moments in alternating up and down directions. In this way, the average continuous rotational symmetry of amorphous materials is broken while the average $C_p T$ symmetry is respected. Another approach is to apply local magnetic fields under the substrate with alternating directions for each 1/p sector during the growing process of amorphous materials. We understand that realizing the phase in experiments is a significant challenge. However, it is theoretically possible.
We therefore believe that our work is novel and is far from a straightforward generalization of Refs. [30,31] and it is also relevant to physical amorphous materials. We have added the discussion on the relevance to physical amorphous materials in the revised manuscript and hope that these responses will satisfy the referee and convince the referee to recommend publication of our work in SciPost Physics.
Comment 2: What guarantees that there are no gapless modes at the boundary of the segments after restoring the rotational symmetry when constructing the Hamiltonians H j, or when applying periodic boundary conditions? As described in the text, the amorphous structure is not completely uncorrelated, rather it is a random set of hard disks. ls this constraint on the minimal interatomic distance obeyed by the symmetrized systems? Based on the description of the construction, it is not, which may result in additional sub-gap states at the gluing interfaces in the symmetrized and periodic boundary condition systems. I ask the authors to clarify these questions and demonstrate that no new subgap physics arises at the gluing interfaces, for example by examining the spatial localization of the low-energy states that appear at large system sizes (fig 3c inset).
Reply: We thank the referee for the nice question. We agree with the referee that when we restore the rotational symmetry, the new system may not satisfy the constraint on the minimal interatomic distance. As a result, the sub-gap states arise in the symmetry restored system as shown in the added Fig. 7(a) in the revised manuscript, where the configuration averaged density distribution of two low-energy states near zero energy under periodic boundary conditions (PBCs) is plotted. However, we would like to clarify the following two points:
-
We only use the symmetry restored system to calculate the topological invariants shown in Fig. 3(a). For all the other results, we consider random lattices without restoring the symmetry. For example, in Fig. 3(c), we calculate the energy gap for random lattices without restoring the symmetry so that no sub-gap physics arises from the gluing operations. This can be clearly seen in the added Fig. 7(b) where the configuration averaged density distribution of two low-energy states near zero energy for the system under PBCs is plotted. Thus, the gapless behavior arises from the bulk states.
-
To calculate the topological invariant in Fig. 3(a), we consider restoring the rotational symmetry. Thus, the topological invariant $\chi$ can be constructed. For the quadrupole moment, the rotational symmetry is not required. One can directly calculate the quadrupole moment for the original system without restoring the symmetry. We find that the quadrupole moment exhibits nonzero values in the topological region and its value increases toward 0.5 with system size (see the added Fig. 7 (c)), similar to the quadrupole moment calculated for the symmetry restored system. In addition, the calculated topological phase transition points agree well with the results of the energy gap and localization properties. All these results suggest that despite the presence of introduced sub-gap states, their existence does not affect the calculation of topological invariants.
We have added three figures in the Appendix C to show the existence of mid-gap states and provide arguments that their existence does not qualitatively affect the calculation of topological invariants there.
Comment 3: Why does the invariant chi_n not depend on n? The manuscript only presents numerical evidence for chi_1=chi_3. I would expect a similar scenario as in ref 28 SM, when additional symmetries forcing a vanishing Chern-number are responsible for this equality, and is not forced by protecting symmetries. It would be worth to check whether this is the case, or at least comment on the possibility.
Reply: We thank the referee for the very interesting question. In the following, we will show that the equality $\chi_1=chi_3$ should not be forced by additional symmetries.
According to Phys. Rev. B 82, 184525 (2010), for a Hamiltonian with particle-hole symmetry, there exists a relation $e^{i\pi C}=\prod_{K_0}\nu(K_0)$, where C is the Chern number and $\nu(K_0)$ is the sign of the Pfaffian of the anti-symmetrized Hamiltonian at particle-hole symmetry momenta $K_0$. In Ref. [30], the equality is generalized to the quasicrystal case with $C_8$ rotational symmetry, reading $e^{i\pi C}=\prod_{n}\nu_n$, where $\nu_n=\nu_{n,0}/\nu_{n,\Pi}$ with $\nu_{n,0}$ ($\nu_{n,\Pi}$) being the sign of the Pfaffian of the anti-symmetrized Hamiltonian restricted in the eigenspace of $C_8 M$ operator with eigenvalue $\omega_n$=e^(in\pi/8)$ for n=[±1,±3,±5,±7] at the high-symmetry momentum 0 ($\Pi$). Due to the zero Chern number of the effective quasicrystalline Hamiltonian, $\prod_n v_n=1$. Besides, the topology protected by $C_4$ symmetry is trivial, so $\chi_1=\chi_7$ and $\chi_3=\chi_5$ for $\chi_n=(1-\nu_n)/2$, which also ensures $\prod_n \nu_n=1$. Similarly, in our amorphous model, we also have $\chi_1=\chi_7$ and $\chi_3=\chi_5$, as discussed in Appendix B.
However, for the quasicrystal model in Ref. [30], $\chi_1\neq \chi_3$. As an indicator of the non-trivial higher-order topology, $\chi_1=\chi_7=1$. In contrast, $\chi_3=\chi_5=0$ because the subspaces with $C_8 M$ eigenvalues $e^(±i3\pi/8)$ and $e^(±i5\pi/8)$ are empty, which is straightforward given the matrix representation of the $C_8 M$ operator $U_(C_8 M)=\sigma_z e^(-i\pi \tau_z/8)=diag(e^(-i\pi/8),e^(i\pi/8),e^(i7\pi/8),e^(-i7\pi/8) ) in the basis of the onsite internal degrees of freedom in Ref. [30]. For the amorphous model in our paper, numerical results also demonstrate $\chi_1 \neq \chi_3$ near the phase transition point in a finite system. The transition points of $\chi_1$ and $\chi_3$ as a function of $m_z$ are different, as shown in the newly added Fig. 6(a). The discrepancy implies that $\chi_1 = \chi_3$ is not protected by any symmetry. However, the difference between the two transition points $\Delta m_z$ exhibits a power-law decay with the increase of the lattice size L as shown in Fig. 6(b), suggesting that $\chi_1=\chi_3$ in the thermodynamic limit. Perhaps this equality in the thermodynamic limit is protected by some emerging symmetry in an infinite system, which indeed deserves future investigations.
We have added the figure of $\chi_1$ and $\chi_3$ as a function of $m_z$ for a typical random configuration in the revised manuscript.
Comment 4: Why is it necessary (or advantageous) to distort the system to calculate the quadrupole moment? Is the quantity measured this way really the quadrupole moment, or perhaps some higher moment? It is really unclear to me how a procedure like this manifestly breaking the protecting rotational symmetries, can extract a topological invariant.
Reply: We thank the referee for the question. To be short, if we do not perform the site transformation, then we always obtain zero quadrupole moment in our case with p=8.
Specifically, similar to the polarization, the quadrupole moment is a widely used topological invariant to characterize the quadrupole topological insulator. The higher-order topology of these insulators can be protected by chiral symmetry or particle-hole symmetry as we have previously proved that chiral symmetry (or particle-hole symmetry) protects the quantization of the quadrupole moment (see Refs. [62,63]). Similar to the Berry phase, the quadrupole moment is a $Z_2$ topological invariant. For a quadrupole insulator on square lattices, when there are odd number of corner modes, the quadrupole moment is equal to 0.5. However, in our case with p=8, there are two corner modes in each quarter in a topological phase, which may explain why the traditional quadrupole moment vanishes. To obtain the reliable topological invariant to characterize the topology of the system, we propose the site transformation while keeping the bulk wave functions unchanged. Such a method indeed gives the correct topological characterization of the system.
In fact, in our another paper (arXiv:2307.14974), we apply the method to calculate the generalized quadrupole moment of a quasicrystalline system. We find that it can correctly characterize the topology of the 2D quasicrystal, similar to the $Z_2$ index based on the rotational symmetry. In the paper, we also generalize the quadrupole moment to characterize the chiral and helical hinge modes in 3D quasicrystals based on the site transformations. For example, one can use the winding number of the generalized quadrupole moment to characterize the chiral mode of 3D second-order topological insulators in quasicrystals without crystalline counterparts. We therefore believe that the generalized quadrupole moment introduced here will play an important role in the study of higher-order topology in non-crystalline systems.
For the physical meaning of the quadrupole moment, let us consider the limiting case of the Benalcazar-Bernevig-Hughes model where the intracell hoppings are zero . In this case, the quadrupole moment is equal to 0.5, and the Wannier center is localized at a maximal Wyckoff position (sorry that the submission system does not allow us to attach a figure to illustrate it). However, in the general case, it remains an open question whether the connection of the nontrivial quadrupole moment to the position of Wannier center is always true. In the non-crystalline case, this connection becomes even more elusive. In our case, this quantity may represent a hidden moment structure. We admit that we do not know its exact nature, but it does serve as a topological invariant. This problem is indeed interesting and merits further study.
We also would like to point out that the rotational symmetry is not an indispensable ingredient for the quadrupole insulator. Without it, a system can still exhibit a quadrupole moment of 0.5 with zero-energy corner modes as long as chiral or particle-hole symmetry persists. To clarify our viewpoint, we consider a 2D quasicrystal with eight-fold rotational symmetry and add some structural disorder to explicitly break the symmetry. We find that this system still has eight zero-energy corner modes and its generalized quadrupole moment is still equal to 0.5. We understand that the rotational symmetry is important as it ensures that the local DOS at zero energy also respects this symmetry. In the amorphous case, the average rotational symmetry plays the role.
We have added some discussions to show that the generalized quadrupole moment can also be used to identify the higher-order topology of 2D and 3D quasicrystals in the revised manuscript.
Comment: The manuscript studies topological phases of amorphous systems protected by chiral and average p-fold (p even) rotation symmetries. The analysis is carried out using a tight-binding model on a random graph, using "momentum-space" invariants derived from twisted periodic boundary conditions, quadrupole moment, and spectral signatures, the manuscript is well written, and the results are sound.
In my opinion the manuscript is an interesting, but fairly straightforward generalization of earlier work on quasicrystalline topological phases with added disorder (see point 1of the Weaknesses section for details).Hence l recommend moving the manuscript to SciPost Physics Core, and publication with minor clarifications: or would request the authors to further support the novelty of the work, and motivate why the paper meets the acceptance criteria of SciPost Physics in light of the questions raised.
Reply: We thank the referee for the nice summary of our work, the positive comment that “The manuscript is well written, and the results are sound” and the critical comments and constructive suggestions. We have follow the referee’s suggestion to clarify the novelty of our work (see our reply to Comment 1) and hope that this considerably improved version will satisfy the referee and convince him/her to recommend publication of this work in SciPost Physics.
Comment 5: Minor change requests below, for major questions see the Weaknesses section.
The hopping Hamiltonian in eqn.1 for p=8 is (up to a basis transformation, and inclusion of bond-length-dependent prefactors) is identical to that of ref (28), and the generalization of the last term for cases with p other than 8 was presented in ref(29), l ask the authors to make this clear by citations when introducing the Hamiltonian.
Reply: We thank the referee for the nice suggestion. We have followed the suggestion to add the sentence that “This Hamiltonian can be derived through a unitary transformation of the Hamiltonian introduced in Refs. [30,31].” in the revised manuscript.
Comment 6: The following statement is unclear to me: “Note that near m_z 0.3 there appears an intermediate region with the coexistence of topologically nontrivial and trivial samples where the gapless bulk states are extended as shown in Fig. 3(d).” Should clarify the text or fig. 3 to make this clear.
Reply: We thank the referee for the comment. We have modified the sentence to “Note that near m_z≈0.3, $\overline{\chi}$ lies between 0 and 1 and does not converge to 0 or 1 as L increases, implying the existence of an intermediate region with the coexistence of topologically nontrivial and trivial samples. In this region, the gapless bulk states are extended as shown in Fig. 3(d).”
Comment 7: Should mention in the caption what the vertical dashed grey lines in fig. 3 denote.
Reply: We thank the referee for the nice suggestion. We have followed the suggestion to add the sentence in the revised caption of Fig. 3 that “The higher-order topological phase is separated from other phases by the vertical dashed grey lines”.
Comment 8: "high-symmetric momenta" should be, following standard terminology, "high-symmetry momenta".
Reply: We thank the referee for the nice suggestion. We have followed the suggestion to change “high-symmetric momenta” to “high-symmetry momenta” in the revised manuscript.

---

## Round 3 · List of Changes

1. We have added two figures in Fig. 3 and the corresponding discussion on the IPR in the revised manuscript.
2. We have added the discussion on the relevance of our model to physical amorphous materials in the revised manuscript.
3. We have added three figures in the Appendix C to show the existence of sub-gap states and provide arguments that their existence does not qualitatively affect the calculation of topological invariants there.
4. We have added the figure of $\chi_1$ and $\chi_3$ as a function of $m_z$ for a typical random configuration in the revised manuscript.
5. We have added some discussions to show that the generalized quadrupole moment can also be used to identify the higher-order topology of 2D and 3D quasicrystals in the revised manuscript.
6. We have added the sentence that “This Hamiltonian can be derived through a unitary transformation of the Hamiltonian introduced in Refs. [30,31].” in the revised manuscript.
7. We have modified the description of the phase with coexisting nontrivial and trivial samples in the revised manuscript.
8. We have added an explanation in the caption of Fig. 3 to clarify the meaning of the vertical dashed grey lines in the revised manuscript.
9. We have changed “high-symmetric momenta” to “high-symmetry momenta” in the revised manuscript.

---

## Editorial Decision

published